# Rapeseed Domestication Affects the Diversity of Rhizosphere Microbiota

**DOI:** 10.3390/microorganisms11030724

**Published:** 2023-03-11

**Authors:** Zhen Zhang, Lu Chang, Xiuxiu Liu, Jing Wang, Xianhong Ge, Jiasen Cheng, Jiatao Xie, Yang Lin, Yanping Fu, Daohong Jiang, Tao Chen

**Affiliations:** 1State Key Laboratory of Agricultural Microbiology, Huazhong Agricultural University, Wuhan 430070, China; 2Hubei Key Laboratory of Plant Pathology, College of Plant Science and Technology, Huazhong Agricultural University, Wuhan 430070, China; 3College of Plant Science and Technology, Huazhong Agricultural University, Wuhan 430070, China

**Keywords:** rapeseed domestication, *Brassica napus*, *Brassica rapa*, *Brassica oleracea*, rhizosphere microbiota, 16S rRNA gene sequencing, synthetic *Brassica napus*

## Abstract

Rhizosphere microbiota is important for plant growth and health. Domestication is a process to select suitable plants to satisfy the needs of humans, which may have great impacts on the interaction between the host and its rhizosphere microbiota. Rapeseed *(Brassica napus*) is an important oilseed crop derived from the hybridization between *Brassica rapa* and *Brassica oleracea* ~7500 years ago. However, variations in rhizosphere microbiota along with rapeseed domestication remain poorly understood. Here, we characterized the composition and structure of the rhizosphere microbiota among diverse rapeseed accessions, including ten *B. napus*, two *B. rapa*, and three *B. oleracea* accessions through bacterial 16S rRNA gene sequencing. *B. napus* exhibited a higher Shannon index and different bacterial relative abundance compared with its wild relatives in rhizosphere microbiota. Moreover, artificial synthetic *B. napus* lines G3D001 and No.2127 showed significantly different rhizosphere microbiota diversity and composition from other *B. napus* accessions and their ancestors. The core rhizosphere microbiota of *B. napus* and its wild relatives was also described. FAPROTAX annotation predicted that the synthetic *B. napus* lines had more abundant pathways related to nitrogen metabolism, and the co-occurrence network results demonstrated that *Rhodoplanes* acted as hub nodes to promote nitrogen metabolism in the synthetic *B. napus* lines. This study provides new insights into the impacts of rapeseed domestication on the diversity and community structure of rhizosphere microbiota, which may highlight the contribution of rhizosphere microbiota to plant health.

## 1. Introduction

Plants possess a diverse but taxonomically structured microbial community, and plant microbiota can colonize every accessible plant tissue [1]. These microbes can form complex associations with plants, playing important roles in promoting plant productivity and health in natural environments [2]. Due to the important role of plant-associated microbes in plant health and disease resistance [3], several studies have explored the composition of microbiota in *Arabidopsis thaliana* [4,5,6,7] and its close relatives [8], barley [9], maize [10,11], rice [12,13], soybean [14], wheat [15], and some other plants [16].

*Brassica napus* (AACC, 2n = 38) originated from the Mediterranean region about 7500 years ago, which was formed from the natural hybridization of two diploid ancestors *Brassica oleracea* (CC, 2n = 18) and *Brassica rapa* (AA, 2n = 20) [17,18]. *B. napus* was domesticated as an important oil plant about 400 years ago. The short evolution and domestication time has resulted in very limited genetic diversity in the *B. napus* genome [19,20,21,22]. Domestication has been demonstrated to affect plant microbiota in common beans and rice [23,24,25], and can lead to highly different microbiota composition of a crop compared with that of its wild ancestor. Particularly, research on rice, sugar beet, barley, *A. thaliana*, and lettuce has indicated that wild species have a higher abundance of *Bacteroidetes* [23,26], which is closely related to plant genotypes and traits [27]. Although *B. napus* has been domesticated for a relatively short period to form only three ecotypes (spring-, winter-, and semi-winter oilseed rape) for adaptation to different vernalization and flowering time [28,29], it remains to be determined whether this short period of domestication has affected its microbiota and whether *B. napus* already has similar characteristics to other domesticated plants.

Reconstruction of a breeding population through interspecific hybridization and genome reconstruction of related species can provide germplasm resources for crop genetic improvement [30,31]. The gene pool of *B. napus* has been expanded by the infiltration of genes from *B. rapa* and synthetic materials resulting from artificial crosses between two diploid ancestors [22,32]. However, the differences in microbiota between the synthetic species and modern domesticated species remain unknown.

In this study, we examined two accessions of *B. rapa*, three accessions of *B. oleracea*, and ten accessions of *B. napus* (two spring-type rapeseed, two winter-type rapeseed, four semi-winter rapeseed, two synthetic *B. napus*, including G3D001 [19] and No.2127 [33]) in terms of rhizosphere microbiota composition and structure to determine the effect of domestication on *B. napus*. The composition, structure, and function of the rhizosphere microbiota of *B. napus* and synthetic *B. napus* were also compared to evaluate the importance of interspecific hybridization in the breeding of *B. napus* from a microbiota perspective.

## 2. Materials and Methods

### 2.1. Plant Cultivation and Soil Collection

Ten *B. napus* accessions (two spring, two winter, four semi-winter rapeseed and two synthetic *B. napus* G3D001 and No.2127), two *B. rapa* accessions (one is green leaves and the other is purple leaves), and three *B. oleracea* accession (Jingfeng 1 is cabbage and the other two are loose head) were used in this study (Appendix A). The seeds of the plants were first surface disinfected in 1.5% sodium hypochlorite solution for 15 min before sowing, and then the disinfected seeds were sown into each pot filled with a uniform substrate and grown in a growth chamber (~20 °C, 60% relative humidity under a 12 h light/12 h dark cycle). Samples were collected at 31 days after germination. The plants were removed from the pots. Loose soil attached to the roots was removed until there was only 2–3 mm of attached soil on the roots. Then, the roots were placed in a 15 mL centrifuge tube with 10 mL PBS buffer, shaken for 30 s, and spun for 10 min using a mixer. The roots were then removed and only the remaining liquid was retained, and the supernatant was removed by centrifugation for the rhizosphere sample [34,35]. Each accession with 8 replicates was collected, and only one *B. rapa* accession (due to a poor germination rate) had 4 replicates—a total of 116 samples were placed in liquid nitrogen for rapid freezing, and then stored in an ultra-low temperature refrigerator at −80 °C.

### 2.2. DNA Extraction, PCR Amplification, and Sequencing

Rhizosphere soil samples were used for bacterial 16S rRNA gene profiling by Illumina sequencing. DNA was extracted from each sample using the OMEGA Soil DNA Kit (D-5635-02) (Omega Bio-Tek, Norcross, GA, USA). Extracted DNA was subjected to 0.8% agarose gel electrophoresis for molecular size determination, and DNA was quantified using Nanodrop. The V5–V7 region of the bacterial 16S rRNA gene was amplified by degenerate PCR primers 799F and 1193R [5,9]. Amplicon sequencing and generation of 250 bp paired-end reads were performed at Personalbio (Shanghai, China) based on the Illumination NovaSeq platform.

### 2.3. Bioinformatics Analysis on 16S rRNA Gene Profiling

Amplicon downstream data were saved in Paired-end (PE) FASTQ format, and the sequencing raw data were checked for quality using the software FastQC (version0.11.9) [36]. The clustering of OTUs was implemented by the software VSEARCH (version 2.8.1) [37]. Firstly, the double-ended data were merged using the --fastq_mergepairs command for the removal of primers and indices from the merged double-ended sequences, which was implemented by the software cutadapt (version 2.9) [38]. Length, quality, and fuzzy base filtering were performed using the command --fastq_filter. Chimera removal was performed by both De novo and Reference database methods. Clustering for OTU was achieved by --cluster_size, and the similarity threshold --id was set to 97%.

Taxonomic annotation of OTUs was achieved by comparing representative sequences of OTUs with representative sequences in the Greengenes database using the q2-feature-classifier command of the qiime2 (version 2019.11) software [39,40,41]. Diversity was calculated by first constructing an evolutionary tree using q2-phylogeny and then computing alpha diversity (Shannon, evenness index, Observed OTU, and Faith’s Phylogenetic) and beta diversity (Bray–Curtis dissimilarity, weight_unifrac) using q2-diversity [42,43]. A generalized linear model with a negative binomial distribution in the R (version 4.0.5) language edgeR package was used to analyze the variance of OTU taxa between each group [44,45]. Visual analysis of box line plots, bar plots, volcano plots, PCoA, extended error bar plots, and ring plots was performed using R language packages such as ggplot2, vegan, amplicon, tidyverse, and patchwork [46,47].

### 2.4. Core Microbial Analysis

For core microbiological analysis, we used the rarefied OTU table (16,000 readings per sample). The OTUs present in all experimental samples were counted as the rhizosphere core microbiota of *B. napus*, *B. oleracea*, and *B. rapa*. The composition of rhizosphere core microbiota at the phylum and genus level was shown using a double-layer pie chart, and boxplots were used to show the distribution of selected genera in *B. napus*, *B. oleracea*, *B. rapa*, and synthetic *B. napus* (G3D001 and No.2127) on the relative abundance. The rhizosphere core microbiota of *B. napus* was evaluated using all samples of all 10 species of *B. napus* and synthetic *B. napus*, and the OTUs that were consistently present in these samples were counted as the core microbiota of *B. napus*.

### 2.5. Rhizosphere Microbiota Functional Prediction and Co-Occurrence Network Analysis

The functions of OTUs in these samples were annotated by FAPROTAX (a database that converts microbial communities into putative functional profiles based on the literature on current culturable strains) using the annotated rarefied OTU table from the Greengenes database as input [41,48]. Heat maps were used to show the functions of nitrogen metabolism-related OTUs associated with nitrogen metabolism and their relative abundance in synthetic *B. napus* (G3D001 and No. 2127) and *B. napus*. Differences in nitrogen metabolism pathways are shown by cluster box line plots.

Co-occurrence network analysis was performed using high-throughput sequencing data to assess interactions between microbial taxa in *B. napus*. Best practices for co-occurrence network construction were strictly followed [49]. OTU counts > 50 and the presence of rarefied OTUs in at least 10 samples were used as input. Non-random co-occurrence analysis was performed by SparCC with 20 iterations and 100 bootstrap samples to infer pseudo-p values in the SpiecEasi r package [50,51]. Co-occurrence networks were constructed using correlations >0.7 or <0.7 (*p* < 0.01). The network was visualized using the ForceAtlas2 layout in Gephi (v0.9.2) software [52]. Gephi was used to calculate the degree and betweenness centrality of the microbial network. Hub OTUs were determined with degree >30 and betweenness centrality >200 as thresholds.

### 2.6. Statistical Analysis

Means were compared on alpha diversity by one-way ANOVA and Tukey’s post hoc test in the agricolae r(v4.1.3) package [53]. In the vegan r(v4.1.3) package [47], PERMANOVA was performed with adonis function with a ‘bray’ method and 9999 times of permutation. Differences in levels of Phylum were compared using the Kruskal–Wallis rank sum test. The comparison of family level and nitrogen metabolism was tested by Welch’s *t*-test and Bonferroni correction.

## 3. Results

### 3.1. Rhizosphere Microbiota Diversity of B. oleracea, B. rapa, and B. napus

To characterize the composition and difference in the rhizosphere microbiota of *B. napus* and its wild relatives, the rhizosphere soil samples of ten *B. napus* accessions (spring, winter, semi-winter rapeseed and two synthetic *B. napus* G3D001 and No.2127), two *B. rapa* accessions, and three *B. oleracea* accession were taken for a comparison analysis (Appendix A). A total of 12,324,260 high-quality sequences were obtained from 116 samples (ranging from 64,168 to 152,242; 106,243 in average). After filtering, denoising, and removal of chimeras and low-read sequences (<20 counts), 1950 bacterial OTUs with 97% sequence similarities or above were identified, which were then rarefied to 16,000 reads per sample for subsequent analysis. After rarefaction, two low-depth samples were removed. Rarefaction curves obtained based on both observed OTUs and Faith PD alpha diversity indices indicated that the majority of members in the root microbiota were captured from each accession in our data (Appendix A).

The Shannon index indicated that the microbiota diversity of *B. napus* was significantly higher than that of its wild relatives *B. oleracea* and *B. rapa* (Figure 1A), as indicated by the more observed OTUs or higher Pielou evenness in *B. napus* than those in *B. oleracea* and *B. rapa* (Appendix A). Principal coordinate analysis (PCoA) based on Bray–Curtis distance revealed that *B. napus* was clearly separated from *B. rapa* on the transverse axis, but partially overlapped with *B. oleracea* (Figure 1B). The *B. oleracea* accessions of G206 and *Brassica incana* were overlapped with *B. napus*, while Jingfeng 1 was separated from *B. napus* (Appendix A). Interestingly, the synthetic *B. napus* G3D001 and No.2127 were distinctly separated from *B. napus* on the abscissa axis (Figure 1B). Permutational multivariate analysis demonstrated that the genotype explained only 9.95% of the total variance between *B. napus* and *B. oleracea*, 18.11% of that between *B. rapa* and *B. napus*, 35.72% of that between G3D001 and *B. napus*, and 27.97% of that between No.2127 and *B. napus* (Appendix A, *p* < 0.001, PERMANOVA by Adonis). A higher proportion of total variance could be explained by genotype for synthetic *B. napus* G3D001 and No. 2127. Similar results were also obtained by PCoA using weighted-UniFrac distances (Figure 1C). These results suggested that *B. napus* has a β-diversity close to that of *B. oleracea* and *B. napus* and synthetic *B. napus* have greater variation in rhizosphere microbiota structure.

### 3.2. Taxonomic Structure of the Rhizosphere Microbiota of B. oleracea, B. rapa, and B. napus

The most abundant phyla in the rhizosphere bacterial community of *B. oleracea*, *B. rapa*, and *B. napus* were Proteobacteria (Alphaproteobacteria, Betaproteobacteria, and Gammaproteobacteria), Bacteroidetes, and Actinobacteria (Figure 1D). *B. rapa* showed a higher relative abundance of Betaproteobacteria (Figure 1D, Appendix A). At the family level, Flavobacteriaceae, Hyphomicrobiaceae, Xanthomonadaceae, Caulobacteriaceae, Burkholderiaceae, Comamonadaceae, Rhizobiaceae, and Pseudomonadaceae were dominant in the rhizosphere microbiota of *B. napus*, *B. oleracea*, and *B. rapa* (Figure 2A,B). *B. rapa* had significantly higher relative abundance of Burkholderiaceae than *B. napus*, while *B. napus* was significantly enriched in Xanthomonadaceae, Comamonadaceae, Sphingomonadaceae, and Weeksellaceae (Figure 2A). *B. oleracea* had higher but not significantly different abundance of Pseudomonadaceae compared with *B. napus*, which could be ascribed to its enrichment only in the *B. oleracea* accession Jingfeng 1 (Appendix A). The rhizosphere microbiota of *B. napus* exhibited significantly higher relative abundance of Comamonadaceae, Rhizobiaceae, and Weeksellaceae than that of *B. oleracea* (Figure 2B). Moreover, compared with synthetic *B. napus* G3D001 and No.2127, *B. napus* had significant enrichment of Flavobacteriaceae in the rhizosphere at the family level, with around 10% of difference in relative abundance (Figure 2C,D). These results indicated that Flavobacteriaceae was significantly depleted in G3D001 and No.2127 relative to *B. napus*, which is the main reason for the difference in rhizosphere microbiota between *B. napus* and the two synthetic *B. napus* lines.

To determine the differential OTUs, edgeR based on the negative binomial distribution was used to analyze the difference in 1950 OTUs in the rhizosphere microbiota of *B. oleracea*, *B. rapa*, and *B. napus*. Compared with those in *B. napus*, 346 OTUs (74.92% *Pseudomonas*) were enriched and 301 OTUs (28.22% *Chryseobacterium* and 20.74% *Flavobacterium*) were depleted in *B. oleracea* (Figure 3A, Appendix A); and 197 OTUs (74.92% *Burkholderia*) were enriched and 163 OTUs (42.25% *Chryseobacterium*) were depleted in *B. rapa* (Figure 3B, Appendix A). The differential OTUs in *B. napus* and synthetic *B. napus* were also analyzed. As a result, 402 OTUs (32.66% *Burkholderia*, 20.04% *Pseudomonas*, and 7.72% *Rhodoplanes*) were enriched and 267 OTUs (27.05% *Flavobacterium* and 12.13% unclassified *Xanthomonadaceae*) were depleted in G3D001 (Figure 3C, Appendix A); while 347 OTUs (18.87% *Pseudomonas*, 13.52% *Burkholderia*, 12.15% *Rhodoplanes*) were enriched and 282 OTUs (24.12% *Flavobacterium* and 12.32% unclassified *Xanthomonadaceae*) were depleted in No.2127 (Figure 3D, Appendix A). Relative to *B. napus*, the number of differential OTUs with higher abundance (log2 relative abundance >10) was 15 in *B. oleracea*, 12 in *B. rapa*, 84 in G3D001, and 84 in No.2127 (Figure 3). These results indicated that the rhizosphere differential OTUs among *B. oleracea*, *B. rapa*, and *B. napus* were mainly related to some rare species, while *B. napus* and synthetic *B. napus* were not only different in rare species but also in some OTUs with high abundance. A significant overlap of differential OTUs was found between G3D001 and No.2127, among which 222 OTUs were co-enriched and 220 OTUs were commonly depleted in the two synthetic *B. napus* lines (Appendix A), indicating that synthetic *B. napus* G3D001 and No.2127 have similar composition of rhizosphere microbiota.

### 3.3. Rhizosphere Core Microbiota Communities of B. napus, B. oleracea, and B. rapa

Among the 1950 OTUs, 82 OTUs were present in the rhizosphere microbiota of all 15 accessions (Figure 4), which accounted for only 4.21% of the total OTU number, but accounted for 70.79% of the total read number. At the phylum level, the core microbiota was mainly composed of 61 *Proteobacteria* OTUs, including 43 *Alphaproteobacteria* OTUs, 9 *Betaproteobacteria* OTUs, 7 *Gammaproteobacteria* OTUs, and 2 *Deltaproteobacteria* OTUs, accounting for 71.94% of the average relative abundance. At the genus level, it was mainly composed of unclassified *Xanthomonadaceae* (four OTUs, 10.94%), *Burkholderia* (one OTU, 7.73%), *Rhodoplanes* (seven OTUs, 7.56%), *Pseudomonas* (one OTU, 6.21%), *Devosia* (four OTUs, 5.36%), *Asticcacaulis* (one OTU, 3.83%), and unclassified *Bradyrhizobiaceae* (two OTUs, 2.81%). *Bacteroidetes* (six OTUs, 20.22%) was second only to *Proteobacteria* in the rhizosphere core microbiota, with *Flavobacterium* (four OTUs, 15.1%) being the most important contributor to the core microbiota. Other bacterial phyla in the core rhizosphere microbiota included *Actinobacteria* (nine OTUs, 4.46%), *Acidobacteria* (two OTUs, 1.24%), and *Verrucomicrobia* (one OTU, 1.03%) (Figure 4). The relative abundance of these core microbes was also diverse in *B. napus*, *B. oleracea*, *B. rapa*, and synthetic *B. napus*. *B. napus*, *B. oleracea*, and *B. rapa* had higher abundance of *Flavobacterium* and unclassified *Bradyrhizobiaceae* than G3D001 and No.2127. *Burkholderia* was more abundant in *B. rapa* and G3D001. *Rhodoplanes* was enriched in G3D001 and No.2127, and *Pseudomonas* was abundant in *B. oleracea* and G3D001. These results were similar to those obtained from differential OTUs (Figure 3). Furthermore, the rhizosphere core microbiota of 10 accessions of *B. napus* was determined. It was found that 103 OTUs were always present in all 79 samples of *B. napus*. These OTUs were mainly composed of *Flavobacterium*, unclassified *Xanthomonadaceae*, *Burkholderia*, *Rhodoplanes*, *Pseudomonas*, *Devosia*, and *Asticcacaulis* (Appendix A).

### 3.4. Synthetic B. napus Rhizosphere Bacterial Communities Have Higher Nitrogen Metabolism Capacity

FAPROTAX was used to annotate the functions of OTUs. As a result, G3D001 and No. 2127 were more abundant in pathways related to nitrogen metabolism (Figure 5A,B, *p* < 0.05, Welch’s *t*-test), including nitrate reduction, nitrogen respiration, and nitrate denitrification and nitrification (Figure 5C). A total of 65 OTUs associated with these nitrogen metabolic pathways were also identified. By comparing these OTUs with the differential OTUs between *B. napus* and synthetic *B. napus*, ten, seven, and three OTUs were found to be enriched in *B. napus*, G3D001, and No. 2127, respectively. Fourteen OTUs were enriched in both G3D001 and No. 2127 (Figure 5D). OTU229 (g_*Rhodoplanes*), OTU1108 (g_*Rhodoplanes*), OTU673 (g_*bradyrhizobium*), OTU674 (g_*Enhydrobacter*), and OTU1185 (g_ *Rhodoplanes*) had higher relative abundance in synthetic *B. napus* than in *B. napus* (Appendix A). *Rhodoplanes* involved in nitrate reduction and denitrification processes [54] was significantly more abundant in synthetic *B. napus* than in *B. napus* (Appendix A), indicating that *Rhodoplanes* is the main factor affecting the differences in nitrogen metabolism.

### 3.5. Co-Occurrence Analysis of B. napus

A co-occurrence network analysis was performed on *B. napus* to evaluate the interaction among rhizosphere microbial taxa. *B. napus* rhizosphere microbial network had 66 nodes and 306 significant correlations (208 positive and 98 negative) (Figure 6A). The network could be divided into two modules, and all connections within the modules were positively correlated, while those between the two modules were negatively correlated (Figure 6A). By comparing the nodes in the network with the differential OTUs of synthetic *B. napus* and *B. napus*, it was found that Module 1 with few nodes was enriched in synthetic *B. napus*, while the OTUs enriched in *B. napus* were in Module 2. Four hub OTUs with high degree and high betweenness centrality were identified, among which OTU711 (g_*Flavobacterium*), OTU388 (g_*Flavobacterium*), and OTU656 (g_*Burkholderia*) were in Module 2, while only one hub OTU229 (g_*Rhodoplanes*) was in Module 1 (Figure 6B). These results indicated that different microbial interaction networks were formed in *B. napus* and synthetic *B. napus*. Moreover, OTU229, which was enriched in synthetic *B. napus*, was a hub microbe in the network, which may transmit the effect to the microbial community via microbe–microbe interactions and play an important role in synthetic *B. napus* lines.

## 4. Discussion

Rapeseed is an economically important oilseed crop in the world, which has a short history of evolution and domestication [16,17]. In this study, we characterized the diversity and structure of rhizosphere microbiota of *B. napus* and its wild relatives. Based on α-diversity, we found that the microbiota associated with the rhizosphere of *B. napus* was more diverse compared to the wild relatives (Figure 1A). This finding is consistent with previous studies, higher microbial diversity in the rhizosphere of modern crops than wild ancestors, and agricultural soil conditions, crop management methods, and host genotype may drive this change. Based on the β- diversity results, *B. napus* and wild relatives are different and it is driven by plant genotype; *B. napus* is close to two accessions of *B. oleracea* (Figure 1B and Appendix A). This result is consistent with the finding that the *B. napus* C subgenome has evolved from the ancestor of *B. oleracea* [18], indicating that the *B. napus* C subgenome plays an important role in controlling rhizosphere microbiota diversity. *B. oleracea* has many subspecies—Jingfeng 1 is cabbage and the other two *B. oleracea* accessions are loose head—as well as having significant differences in relative abundance of *Pseudomonas* from *B. napus* (Appendix A).

Domestication is a factor that determines changes in the composition of the plant microbiome. Although the rhizosphere microbiome of both *B. napus* and its wild relatives were dominated by similar bacterial phyla (*Proteobacteria*, *Bacteroidetes*, and *Actinobacteria*), they still have obvious differences between *B. napus* and its wild relatives. Compared with *B. napus*, *B. oleracea* is enriched for *Pseudomonas* and *B. rapa* is enriched for *Burkholderia*, respectively. The *Burkholderia* and *Pseudomonas* group comprises several etiological agents of plant diseases and plant-beneficial and symbiotic species to promote plant physiology and growth [55,56,57,58,59,60]. Some *Pseudomonas* and *Burkholderia* species are also considered as biocontrol agents for various fungi [56,58,61]. Rahman and colleagues [62] demonstrated that a *Pseudomonas* strain isolated from barley seeds has beneficial effects on the host, especially under harsh environmental conditions. This evidence indicates that wild relatives, often living under stressed conditions, can be supported by *Pseudomonas* and *Burkholderia* to cope with abiotic and biotic stresses.

Heteropolyploid plants constitute a new breeding population by reconstructing the genome of their homologous species, which will significantly promote the genetic diversity of the species and the evolution and recombination genome [19,63,64,65]. The breeding of *B. napus* using the genomics and modern breeding techniques to reconstruct the genome would facilitate genome evolution and crop improvement in a short period of time and is far more efficient than domestication that usually requires thousands of years [19]. To investigate whether there are differences between rhizosphere microbiota of domesticated and synthetic plants, we examined the rhizosphere microbiota of *B. napus* and synthetic *B. napus* G3D001 and No.2127. PCoA revealed a clear separation between *B. napus* and synthetic *B. napus* lines (Figure 1B), and differences in the composition of rhizosphere microbiota, including *Flavobacterium, Chryseobacterium, Pseudomonas, Burkholderia*, and *Rhodoplanes* (Appendix A). These differences are much greater than those in rhizosphere microbiota between *B. napus*, *B. rapa*, and *B. oleracea*. These results indicate that artificial genome reconstruction has a greater effect on the rhizosphere microbiota of *B. napus* than domestication.

More abundant *Bacteroidetes* OTUs of wild species have been reported in the seed microbiota of rice and rhizosphere bacterial communities of beet, *Arabidopsis*, barley, and lettuce [24,27]. However, *B. napus* showed no significant difference in the abundance of *Bacteroidetes* from *B. oleracea* and *B. rapa.* Interestingly, the abundance of *Bacteroidetes* in *B. napus* was significantly higher than that in synthetic *B. napus* lines (Appendix A). This may indicate the higher degree of domestication of synthetic *B. napus* than *B. napus*. Furthermore, we found that the rhizosphere microbiota of synthetic *B. napus* was mainly enriched in *Pseudomonas* and *Burkholderia* compared to *B. napus* and depleted *Flavobacterium*. This indicates that the synthetic *B. napus* obtained by artificial interspecific cross breeding is able to inherit the inter-root microbiota of both parents.

*B. napus* is a species with a low nitrogen use efficiency [66,67,68]. Some studies have shown that artificial interspecific hybridization can produce *B. napus* with high nitrogen use efficiencies [69]. It has also been demonstrated that the level of nitrogen use efficiency of the plant host affects the nitrogen metabolism capacity of the plant microbiota [70]. In this study, synthetic *B. napus* lines showed a higher nitrogen metabolism capacity of rhizosphere microbiota than *B. napus* (Figure 5A,B). *Rhodoplanes* enriched in synthetic *B. napus* play an important role in the nitrogen cycle. Co-occurrence network analysis suggested that the microbial network formed two modules corresponding to *B. napus* and synthetic *B. napus*, respectively, in which different central nodes were observed. These results indicate the formation of different microbial interaction patterns between *B. napus* and synthetic *B. napus*. Moreover, *Rhodoplanes* acts as network hub nodes in *B. napus* and was enriched in synthetic *B. napus*. Previous research has indicated that *Rhodoplanes* has a positive correlation with available soil N and can be involved in multiple steps in N cycling, including N fixation, nitrate reduction, and denitrification; furthermore, it may help to improve N availability by conservation tillage and subsoiling [54,71,72]. These facts may be responsible for the differences in the nitrogen metabolism of the rhizosphere microbiota between *B. napus* and synthetic *B. napus*.

## 5. Conclusions

In this study, we used *B. napus*, *B. oleracea*, *B. rapa*, and synthetic *B. napus* as models to study the effects of domestication and interspecific cross breeding on the rhizosphere microbiota of *B. napus*. We found that domestication and interspecific cross breeding significantly affected the diversity and composition of the rhizosphere microbiota of *B. napus*. Moreover, the rhizosphere microbiota of synthetic *B. napus* can inherit some beneficial microorganisms from its parents, and the abundance of nitrogen metabolism-related pathways of the rhizosphere microbiota of synthetic *B. napus* is higher than that of *B. napus* due to the presence of *Rhodoplanes* in the rhizosphere microbiota. This showed that interspecific hybrid breeding can modify the rhizosphere microbiota of *B. napus*. Our work indicated the effects of domestication on the rhizosphere microbiota of *B. napus* and provided insights into the effects of interspecific cross breeding on plants in terms of the microbiota.

## Figures and Tables

**Figure 1 microorganisms-11-00724-f001:**
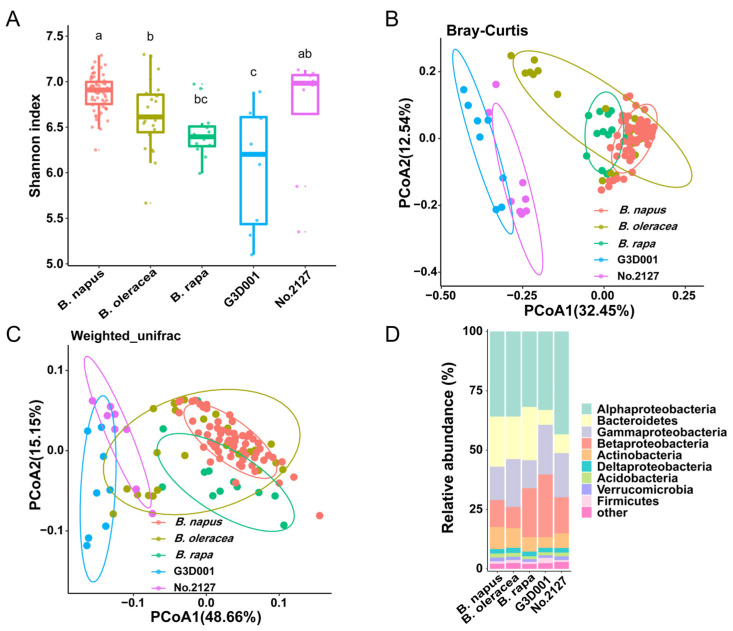
Rhizosphere microbiota of *B. napus*, *B. oleracea*, *B. rapa*, and synthetic *B. napus* G3D001 and No.2127. Shannon index (**A**), unconstrained PCoA based on Bray–Curtis distance. Shannon index indicated by different letters (a, b and c) are significantly different (ab is not significantly different from a or b, bc is not significantly different from b or c). (**B**), unconstrained PCoA based on weight-UniFrac distance (**C**), phylum level distribution (**D**) of the rhizosphere microbiota of *B. napus*, *B. oleracea*, *B. rapa*, and synthetic *B. napus* lines. The numbers of replicated samples in this figure are as follows: *B. napus* (n = 63), *B. oleracea* (n = 23), *B. rapa* (n = 12), G3D001 (n = 8), No.2127 (n = 8). Genotype explained 37.82% of the total variance in these plants (*p* <  0.001, PERMANOVA by Adonis), and ellipses cover 80% of the data for each plant in (**B**,**C**). Proteobacteria is shown in (**D**) at the class level.

**Figure 2 microorganisms-11-00724-f002:**
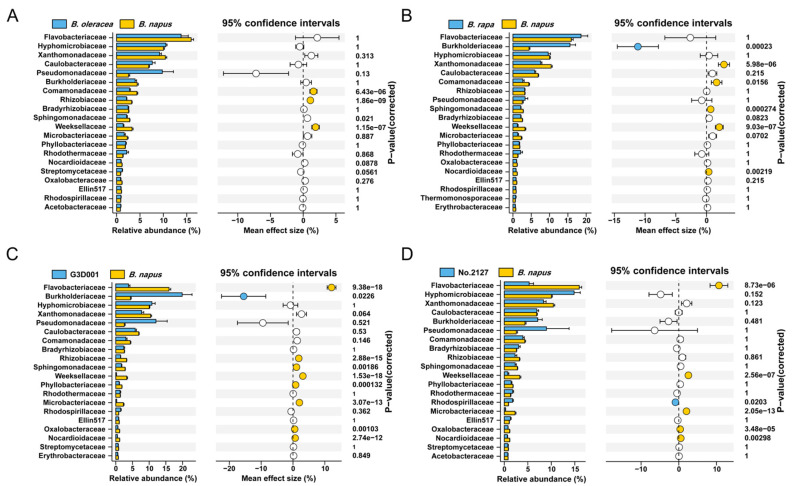
Differential abundance at the family level in *B. napus*, *B. oleracea*, *B. rapa*, and synthetic *B. napus* G3D001 and No.2127. Welch’s *t*-test and Bonferroni correction were performed between the rhizosphere microbiota of *B. napus* and *B. rapa* (**A**), *B. napus* and *B. oleracea* (**B**), *B. napus* and G3D001 (**C**), *B. napus* and No.2127 (**D**) at the family level, respectively. Each vertical bar represents the standard error. The white color circle represents not significant, yellow represents enrichment in *B. napus*, blue represents enrichment in *B. oleracea* (**A**), *B. rapa* (**B**), G3D001 (**C**), No.2127 (**D**), respectively.

**Figure 3 microorganisms-11-00724-f003:**
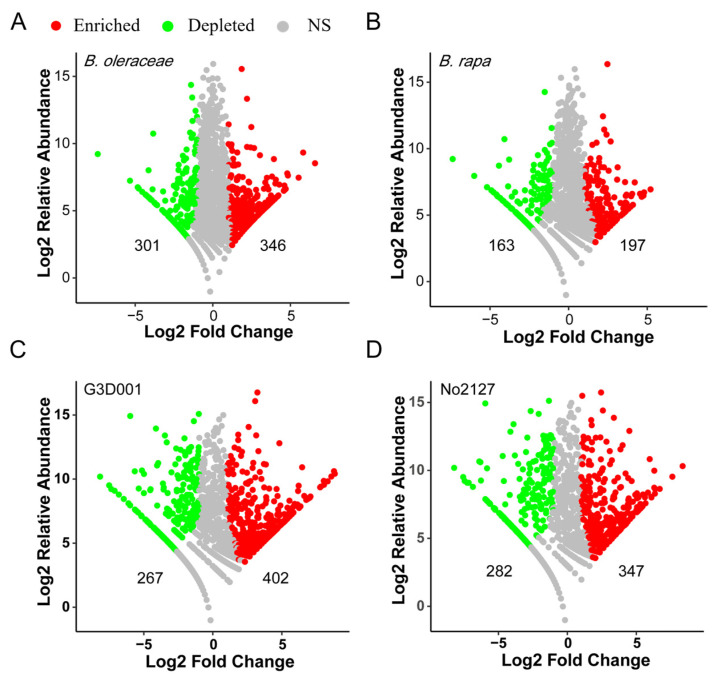
Enriched and depleted OTUs in *B. oleracea* (**A**), *B. rapa* (**B**), and synthetic *B. napus* G3D001 (**C**) and No.2127 (**D**) relative to those in *B. napus.* Each point represents an individual OTU, and the position along the y-axis represents the change in average relative abundance. Numbers represent the number of enriched and depleted OTUs.

**Figure 4 microorganisms-11-00724-f004:**
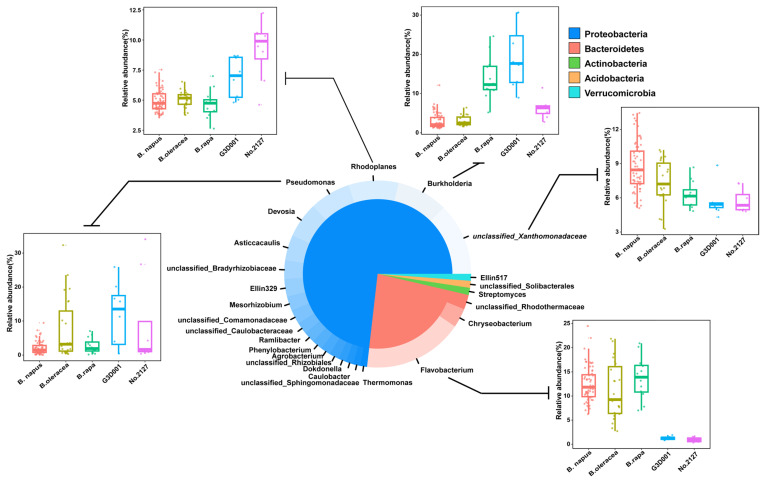
Rhizosphere core microbiota of *B. napus* and its relatives *B. oleracea* and *B. rapa.* Different portions within the inner pie chart represent the bacterial phyla that are part of the *B. napus*, *B. oleracea*, and *B. rapa* core microbiota. The outer donut plot represents the genera that are part of the core, and each genus is assigned to the phylum to which it belongs. Different sizes of the pie and donut portions represent the contribution of each phylum/genus to the total relative abundance. Box plots depict the relative abundance of selected genera.

**Figure 5 microorganisms-11-00724-f005:**
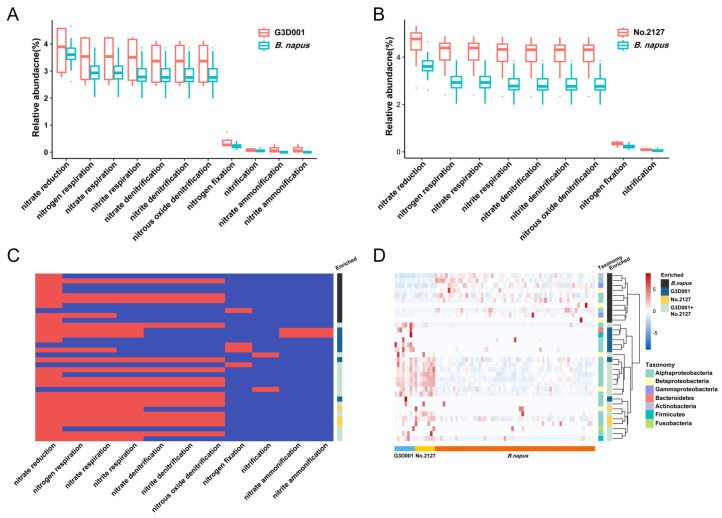
Difference and taxonomy of nitrogen metabolism-related OTUs in the rhizosphere microbiota of *B. napus* and synthetic *B. napus* G3D001 and No.2127. (**A**,**B**) Differences in nitrogen metabolism function of synthetic *B. napus* G3D001 (**A**), No.2127 (**B**) and *B. napus* rhizosphere microbiota, respectively. (**C**) Nitrogen metabolism functions of OTUs enriched in *B. napus* and synthetic *B. napus* based on FAPROTAX. Each row represents an OTU. The presence of functions is shown in red. (**D**) The heat map shows the relative abundance and taxonomy of OTUs enriched in *B. napus* or synthetic *B. napus* G3D001 and No.2127. The presence of functions is shown in red, and the absence of functions is shown in blue.

**Figure 6 microorganisms-11-00724-f006:**
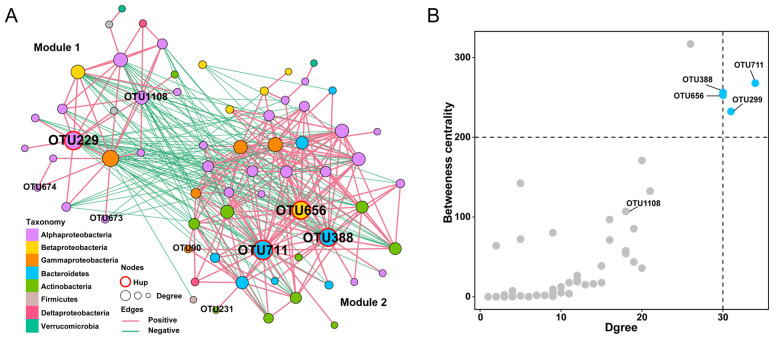
Microbial network of *B. napus* rhizosphere microbiota. (**A**) Co-occurrence network of *B. napus* is based on 79 samples of *B. napus* including synthetic lines. Each node corresponds to an OTU, and edges between nodes correspond to positive (pink) or negative (green) correlations inferred from OTU abundance profiles using the SparCC method (pseudo *p* < 0.01, correlation values < −0.7 or > 0.7). OTUs belonging to different microbial phyla have different colors. Proteobacteria is shown at the class level. The node size reflects their degree in the *B. napus* rhizosphere. (**B**) Hub OTUs of *B. napus.* The dotted line indicates the threshold of hub OTUs.

## Data Availability

All sequencing data used in this study are available in the NCBI under the accession number PRJNA934188.

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
