# Peer review of "Rapeseed Domestication Affects the Diversity of Rhizosphere Microbiota"

_microorganisms, 2023, doi:10.3390/microorganisms11030724_

Round 1
Reviewer 1 Report
This manuscript demonstrated that rapeseed domestication affects the diversity of rhizosphere microbiota using 16S rRNA gene sequencing. The results are interesting, yet the interpretation did not provide any mechanistic understanding. Please rewrite the Discussion section.
Lines 70-80, please provide more details of the experiment design. How many treatments? How many replicates? How were the treatments assigned? How many samples in total?
Line 91-99, how many samples were left after quality control?
Line 112, how many samples were left after rarefaction? What was the threshold? How was the threshold chosen?
Lines 318-365, the interpretation of the results is too general. Please be more specific about how the microbiota was different among different plants and why. What are the mechanisms? What are the implications for ecological functions?
Lines 367-372, please provide more conclusive information rather than repeat the results.
Author Response
Lines 70-80, please provide more details of the experiment design. How many treatments? How many replicates? How were the treatments assigned? How many samples in total?
RESPONSE: Thank you for your suggestion, we modified. Please to see Line 70-73, Line 82-84. Ten B. napus accessions (two spring, two winter, four semi-winter rapeseed and two synthetic B. napus G3D001 and No.2127), two B. rapa accessions (one is green leaf and the other is purple leaf), and three B. oleracea accession (Jingfeng 1 is cabbage and other two are loose head) were used in this study (Supplementary Table S1). The plants grown in the growth chamber. Each accession with 8 replicates were collected, only one B. rapa since poor germination rate is 4 replicates, a total of 116 samples for 16S sequencing.
Line 91-99, how many samples were left after quality control?
RESPONSE: Sequencing raw data were checked for quality using the software FastQC, and the quality of all the 116 sample is good, we did not remove any sample.
Line 112, how many samples were left after rarefaction? What was the threshold? How was the threshold chosen?
RESPONSE: We used 16000 readings per sample for rarefaction, two low-depth samples were removed. The threshold is 16000, we chosen it that are equal to or less than the number of in the smallest sample. We added it. Please to see Lines 161-162.
Lines 318-365, the interpretation of the results is too general. Please be more specific about how the microbiota was different among different plants and why. What are the mechanisms? What are the implications for ecological functions?
RESPONSE: Thank you for your suggestions. We revised. Please to see lines 324-330, 334-335, 337-349, 357-359 and 370-375.
Lines 367-372, please provide more conclusive information rather than repeat the results.
RESPONSE: Thank you for your suggestions. We revised. Please to see lines 399-410.
Reviewer 2 Report
The manuscript is very interesting presenting the novelty of characterizing rhizosphere microbiota of B. oleracea, B. rapa, and B. napus and further comparison with synthetic B. napus lines. Some minor corrections to be made in the manuscript:
Line 242: OUT is misspelled, change to OTU
Line 294: subsection 3.5 is named same as 3.3, did you mean co-occurrence analysis of B. napus?
Line 336: change “Here, we studied the rhizosphere microbiota of B. napus and synthetic B. napus G3D001 and No.2127” to “To investigate whether there are differences between rhizoshpere microbiota of domesticated and synthetic plants we examined the rhizosphere microbiota of B. napus and synthetic B. napus G3D001 and No.2127”
Line 372: change “abundance of Rhodoplanes” to “presence of Rhodoplanes in rhizosphere microbiota”
Author Response
The manuscript is very interesting presenting the novelty of characterizing rhizosphere microbiota of B. oleracea, B. rapa, and B. napus and further comparison with synthetic B. napus lines. Some minor corrections to be made in the manuscript:
Line 242: OUT is misspelled, change to OTU
RESPONSE: Thank you. Revised. Please to see Line 243.
Line 294: subsection 3.5 is named same as 3.3, did you mean co-occurrence analysis of B. napus?
RESPONSE: Thank you. We made a mistake. Revised. Please to see Lines 296-297.
Line 336: change “Here, we studied the rhizosphere microbiota of B. napus and synthetic B. napus G3D001 and No.2127” to “To investigate whether there are differences between rhizoshpere microbiota of domesticated and synthetic plants we examined the rhizosphere microbiota of B. napus and synthetic B. napus G3D001 and No.2127”
RESPONSE: Thank you. Revised. Please to see Lines 357-359.
Line 372: change “abundance of Rhodoplanes” to “presence of Rhodoplanes in rhizosphere microbiota”
RESPONSE: Thank you. Revised. Please to see Line 406-407.
Round 2
Reviewer 1 Report
Agree to publish.